# A Novel Method for Sea-Land Clutter Separation Using Regularized Randomized and Kernel Ridge Neural Networks

**DOI:** 10.3390/s20226491

**Published:** 2020-11-13

**Authors:** Le Zhang, Jeyan Thiyagalingam, Anke Xue, Shuwen Xu

**Affiliations:** 1Key Lab for IOT and Information Fusion Technology of Zhejiang, Hangzhou Dianzi University, Hangzhou 310018, China; akxue@hdu.edu.cn; 2Scientific Computing Department, Science and Technology Facilities Council, Rutherford Appleton Laboratory, Didcot OX11 0QX, UK; t.jeyan@stfc.ac.uk; 3National Lab of Radar Signal Processing, Xidian University, Xi’an 710126, China; swxu@mail.xidian.edu.cn

**Keywords:** radar clutter classification, ECAV based feature extraction, KRR and RRNN, efficient and generalizing

## Abstract

Classification of clutter, especially in the context of shore based radars, plays a crucial role in several applications. However, the task of distinguishing and classifying the sea clutter from land clutter has been historically performed using clutter models and/or coastal maps. In this paper, we propose two machine learning, particularly neural network, based approaches for sea-land clutter separation, namely the regularized randomized neural network (RRNN) and the kernel ridge regression neural network (KRR). We use a number of features, such as energy variation, discrete signal amplitude change frequency, autocorrelation performance, and other statistical characteristics of the respective clutter distributions, to improve the performance of the classification. Our evaluation based on a unique mixed dataset, which is comprised of partially synthetic clutter data for land and real clutter data from sea, offers improved classification accuracy. More specifically, the RRNN and KRR methods offer 98.50% and 98.75% accuracy, outperforming the conventional support vector machine and extreme learning based solutions.

## 1. Introduction

In the context of target tracking and target identification, radar returns from unwanted objects, termed as clutter, compete with the intended true returns. The exact candidates for clutter may vary depending on the context. For instance, when ground objects are of interest, radar returns from nearby sea, waves, and targets in the sea can be considered as clutter. However, when tracking sea-borne targets, returns from ground objects can be considered as clutter. Regardless of the origin of the clutter, their variability is often substantially high so as to render the tracking and identification of targets a difficult process. This difficulty in tracking and identification of targets becomes even more severely challenging when dealing with multiple clutter types. This scenario is very common at and near the interfaces of multiple surface types, such as air and land, air and sea, or land and sea, where the actual targets of concern are bound only to a single surface. For example, a ground based radar should focus only on ground based targets, and thus, returns from sea-bound targets should be treated as clutter. For this reason, discriminating between different clutter types is crucially vital for robust tracking and identification of objects from one surface type [1]. In this paper, we focus on deriving a novel, machine learning method for discriminating sea-land clutter.

There are a number of methods that have been proposed to address this issue, with varying degrees of success. Perhaps one of the earliest and simplest approaches for labeling the land based returns (and thus, to discriminate against sea based returns) is to rely on pre-loaded maps of the land under surveillance [2]. However, such an approach is not always feasible. For instance, a clear land map may not be available, or may not be accurate enough, or the resolution is not sufficient enough to separate the land-sea boundaries. Although manual processing is a possible avenue to address this issue, it is not a scalable technique. A number of model based approaches have been developed to address this issue. These include amplitude distribution analysis [3,4,5,6,7], spectral analysis, auto-regressive modeling [8,9,10], and parametric and nonparametric statistical decision rules. For instance, in [3], a generalized, compound distribution model incorporating thermal noise was used to describe the high resolution radar clutter. In [4], the model parameters were described using generalized compound probability density functions, and these distribution functions were then used to classify the clutter types, in particular for high resolution radar echoes.

Although model-driven approaches have been demonstrated to be significant enough, the accuracy of the land-sea clutter discrimination from a model-driven approach directly depends on the correctness of the underlying model. For instance, this requires the model parameters not only to be captured properly; their range of values also has to be reasonable enough to render accurate results. As such, even if model-driven approaches are considered to be good enough, they do demand a great deal of effort towards the estimation of the relevant parameter space for accurate discrimination of the land-sea clutter. This is one of the major weaknesses of the model-driven approaches.

Another school of thought is to utilize data-driven, machine learning based approaches, in particular supervised learning approaches. Although these approaches can stem from one or more basis models, the actual model is constructed by learning from example data, which are often referred to as training data. Here, the training data are constructed by combining a large number of input and expected output pairs. These pairs are often known as labeled data. The general expectation here is to learn from training so as to have a model that evolves to work on unlabeled data. In other words, the learning process is guided and therefore known as supervised learning. More specifically, the input will be a clutter, and the expected output is the indication of the class under which the input clutter data fall. This approach, however, requires datasets that are somehow labeled. The overall performance is then evaluated using the number of misclassifications or correct classifications.

Among different supervised learning approaches, neural network (NN) or artificial neural network (ANN) based approaches have been shown to be very effective [2,5,6,8,10,11]. The common theme among all the ANN based approaches is that a feed-forward, multi-layer neural network is used to learn to classify a given clutter among multiple classes. The exact mechanism and the overall performance, however, vary based on the network configuration (such as the number of layers, activation functions, and/or neurons per layer) and the number of classes of the clutter. In [12], a system consisting of a feature extractor using wavelet packets and a two-layer back-propagation neural network was used to classify the underwater mine-like targets and non-targets from the acoustic back-scattered signals. In [13], the wavelet method was applied to extract the sea clutter, so that the estimation of oceanographic parameters and target detection were improved. Finally, in [14], the extreme learning machine (ELM), which is a variant of regular ANNs, was used to classify the target and sea clutter. In order to facilitate the classification process, a number of hand-picked features, namely the de-correlation time, K distribution parameters, and the Hurst exponent in the fractional Fourier transform domain, were used as feature vectors.

A number of non-neural network based supervised machine learning approaches have also been used to address this problem. For instance, in [15], the problem of discriminating small maritime targets against sea-clutter was discussed using K nearest neighbor (KNN) and support vector machines (SVMs). A single class based supervised learning approach based on the one-class support vector machine (OCSVM) was also used in the context of clutter discrimination. For instance, in [16], the OCSVM was used for suppressing clutter arising from ocean waves, more specifically in the context of high frequency ground wave radar. Another example is [17], where OCSVM was used to detect ship wakes using active sonar systems. In [18], another variant of SVM, cost sensitive support vector machine (2C-SVM), was used to handle clutter from multiple models of clutter, in particular the Gaussian and K-distributed clutters, and approximate the Neyman–Pearson detector for a specific false alarm rate. More recently, in [19], a KNN classifier was proposed to yield a pixel-level sea-land segmentation of the scene. The different solution types mentioned above are compared in Table 1.

In this paper, we propose two novel techniques for discriminating and classifying the sea and land clutters using two supervised machine learning techniques, namely the regularized randomized neural network (RRNN) and kernel ridge regression (KRR). Although our approaches are supervised learning based, the overall focus here is the classification of clutter as opposed to suppression of clutter or direct target discrimination [2,9,14,15,16,17,18,20]. As such, we use all possible information towards the classification process rather than discarding useful information in favor of target discrimination. Furthermore, the key challenges of distinguishing and classifying the sea-land clutter revolve around high quality features that can provide this aid in the classification between these two classes of clutter. To render the classification process more meaningful, we pay special attention to the feature space, on which the training process relies. In particular, the proposed feature space for the sea and land clutters includes classic parameters covering the statistical characteristics of clutters, short time energy, amplitude variation, and the frequency and properties around autocorrelation. The feature space we propose here, to the best of our knowledge, is novel and different from that of [2,3,4,5,6]. In doing these, we make the following contributions:We propose a novel set of features for improving the classification performance, namely the energy, zero-crossing rate, the autocorrelation, and the shape parameter *V* (ECAV).We present two novel classification systems based on the RRNN and KRR utilizing the ECAV feature space; andWe conduct an in-depth, exhaustive analysis of the proposed methods and compare against the routine baselines, namely the SVM and ELM, to observe the generalization and robustness of the proposed system.

The rest of the paper is organized as follows: In Section 2, we provide the essential background surrounding clutter models and the RRNN and KRR techniques. We then introduce our proposed techniques in Section 3. In Section 4, the experimental results are discussed, and the complexity of the proposed algorithm is analyzed. We finally conclude the paper in Section 5.

## 2. Background

### 2.1. Clutter Modeling

Among many different approaches for modeling clutter [21], modeling them using a statistical (or probabilistic) distribution is a robust and proven technique. Example clutter models include Rayleigh, log-normal, Weibull, and K distributions [6,22].

In general, if a radar looks at a natural surface with a low resolution and at a not very low grazing angle so that no dominant shadowing and multi-path effects are experienced, the resultant clutter distribution is very likely to model the Rayleigh distribution. However, if the resolution cell becomes very small or if illuminated areas contain discrete man-made scatters, the exact number of effective scatters may decrease. With this, the distribution usually becomes much wider than the Rayleigh distribution. Furthermore, if the low grazing angle scatters, multi-path effects, and shadowing are accounted for, the amplitude statistics deviate from the Rayleigh distribution. In these cases, the log-normal, Weibull, and K distributions are the most commonly used distributions to approximate the resulting clutter distribution [6,22].

With this, the amplitude probability density function of a Weibull distribution is:(1)$$p\left(x\right)=\alpha \delta {\left(x\right)}^{\alpha -1}\exp\left(-\delta x^{\alpha }\right),x\geq 0,\alpha ,\delta >0$$
where δ is the scale parameter and α is the shape parameter of the statistic model, which are usually used to model the sea, weather, and land clutter.

The amplitude probability density function of the K distribution is:(2)$$p\left(x\right)=\frac{2\beta }{\Gamma (v)}{\left(\frac{\beta x}{2}\right)}^{v}K_{v-s}\left(\beta x\right),x\geq 0,\beta ,v>0$$
where Γ(·) is the gamma function and $K_{v-s}\left(\cdot \right)$ is the *S*-order modified Bessel function. The model is characterized by the scale parameter β and shape parameter *v*. It is worth noting that the shape parameter is the measure of clutter spikiness. In other words, the clutter becomes spikier with *v* decreasing and becomes Gaussian distributed when *v* approaches infinity. The K distribution describes the clutter as a compound distribution, which consists of two components: the speckle and the local power. It is the particular case of the compound Gaussian (CG) model.

### 2.2. RRNN and KRR

Feed forward neural networks have been extensively used in many domains, primarily for their ability to approximate complex non-linear functions. More specifically, feed forward neural networks are a popular choice for classification tasks. However, there are several aspects that determine the overall performance, which is often judged by the accuracy of the classification. First, the training process, and thus the resulting classification performance on data other than training, relies on the amount of training data used for training, which, in our case, are a scarce resource. Secondly, the overall network performance depends on the optimal selection of the network parameters, which are often referred to as hyperparameters. The parameters for optimal tuning include the number of layers, the activation functions of the layers, the learning rate, the number of neurons in each layer, the loss function used to quantify the convergence, and the optimization function used to minimize the loss function. This is particularly a problem when the dimensionality of the hyperparameter space is large. A large body of work that investigates the hyperparameter optimization problem can be found in [23,24].

The RRNN is, however, a feed forward neural network with a single hidden layer. The weights of the hidden layer are chosen randomly, as opposed to relying on back-propagation or gradient descent, and the output layer is trained by a simple linear learning rule. This model reduces the dimensionality of the hyperparameter space, and the process of finding the optimal values for the hyperparameters becomes rather simplified. In other words, given an RRNN and a feature space, the network is trained by randomized selection of features during a single cycle of training. In particular, a nonlinear squashing function is used to compute the weighted sum of the inputs and to map the input data into the feature space. This does not imply that the parameters describing the hidden nodes, which are independent of the training data, are unconstrained. The parameters for the hidden nodes are randomly generated by adhering to a given continuous probability distribution function.

This approach, outlined and discussed in [25], is different from multi-layer based neural networks and offers excellent efficiency. Once the input weights and biases are chosen during the training, the overall output weight (of the output layer) setup can be considered as a linear system. Furthermore, the output weights (linking the hidden layer to the output layer) can analytically be determined through a simple generalized inverse operation of the hidden layer output matrices.

Consider a basic RRNN model with *L* neurons in the hidden layer and a training set with *N* arbitrary distinct samples of $\left(x_{j},t_{j}\right)$. The output function for the network can then be expressed as [25]:(3)$$\begin{array}{ccc} f_{L}\left(x\right) & = & \sum_{i=1}^{L}\beta_{i}g_{i}\left(w_{i}x_{j}+b_{i}\right)\;,\forall j\in \left\{1\ldots N\right\} \end{array}$$
where $x_{j}$, $w_{i}$, and $\beta_{i}$ are the input, input-to-hidden weight, and hidden-to-output weight vectors, respectively. Furthermore, g(.) and $b_{i}$ are the nonlinear piece-wise activation function and the bias of a hidden node *i*.

Letting $h_{i}\left(x\right)=g_{i}\left(w_{i}x_{j}+b_{i}\right)$ and:$$h\left(x\right)=[h_{1}\left(x\right),h_{2}\left(x\right)\ldots ,h_{L}\left(x\right)]$$
we render a nonlinear feature mapping between the feature space and the outputs. While the input weights are expressed between a pair of input and hidden nodes, the output weights connect the *i*-th hidden node to the output nodes. Thus, Equation (Equation 3) becomes:(4)$$\begin{array}{ccc} f_{L}\left(x\right) & = & \sum_{i=1}^{L}\beta_{i}h_{i}\left(x\right)\\ & = & h(x)\beta \end{array}$$

The output weights vector β should be determined by minimizing the $L_{2}$ norm. In other words,
(5)$$e^{2}=\min_{\beta \in R^{L\times m}}\|H\beta -T{\|}_{2}^{2}$$
where *T* is the target matrix of the training dataset and H is the output matrix of hidden layers, given by:(6)$$\begin{array}{ccc} H & = & \left[\begin{array}{c} h(x_{1})\\ \vdots \\ h(x_{N}) \end{array}\right]\\ & = & {\left[\begin{array}{ccc} g(w_{1}\cdot x_{1}+b_{1}) & \cdots & g(w_{L}\cdot x_{1}+b_{L})\\ \vdots & \vdots & \vdots \\ g(w_{1}\cdot x_{N}+b_{1}) & \cdots & g(w_{L}\cdot x_{N}+b_{L}) \end{array}\right]}_{N\times L} \end{array}$$

The approximation of β can be given by $\beta^{{}^{*}}=H^{†}T$, where $H^{†}$ is the Moore–Penrose generalized inverse of H. In order to obtain better generalization performance of the learning network, the smallest training error and the smallest norm of the output weights must be sought as in [25], i.e.,
(7)$$\min_{\beta }\frac{1}{2}{∥\beta ∥}^{2}+\frac{C}{2}{∥T-H\beta ∥}^{2}$$
where *C* is the regularization parameter between the training error and the norm of the output weights. If N≥L (i.e., H has more rows than columns), then the approximated value of β can be given by:(8)$$\beta^{*}\approx {\left(H^{T}H+\frac{I}{C}\right)}^{-1}H^{T}T$$
where I is an identity matrix of dimension *L*. Conversely, if N<L (i.e., H has more columns than rows), then the approximated value of β can be given by:(9)$$\beta^{*}\approx H^{T}{\left(HH^{T}+\frac{I}{C}\right)}^{-1}T$$
where I is an identity matrix of dimension *N*.

Ridge regression is one of the classical statistical algorithms, and it is often used to transform a non-linear regression function in the original input space as a linear regression function in the feature space. Hence, the dual version of the ridge regression algorithm allows the use of kernel functions. Furthermore, the ridge regression is easy to generalize to KRR using the kernel technique. Assume that we have a training set $\left\{\left(x_{1},y_{1}\right),\ldots ,\left(x_{N},y_{N}\right)\right\}$, with *N* samples. Furthermore, $x_{i}\in R^{n}$, $y_{i}\in R$, for i∈{1,…,N}. For a linear system y=ωx, ridge regression minimizes the following cost [26]:(10)$$J\left(\omega \right)=\lambda {∥\omega ∥}^{2}+\sum_{i}{\left(y_{i}-\omega^{T}x_{i}\right)}^{2}$$
where λ>0 and fixed. Furthermore, λ controls the trade-off between the bias and variance of the estimate. The prediction $\hat{y}$ given by the ridge regression on a new unlabeled data point *x* is [26]:(11)$$\hat{y}=y^{T}{\left(K+\lambda I\right)}^{-1}\kappa$$
where K is matrix of dot products of the vectors, and κ is vector of dot products of *x* and vectors of the training set such that:(12)$$K=\{K\left(x_{i},x_{j}\right)\}$$
and
$$\begin{array}{c} \kappa =\{K\left(x_{i},x\right)\} \end{array}$$
for i,j={1,2,…,N}. In fact, these are also the kernel functions of interest. When using feature vectors, each data point $x_{i}$ is replaced by a corresponding feature vector $\Phi (x_{i})$. Thus,
(13)$$\begin{array}{ccc} K(x_{i},x_{j}) & = & <\Phi \left(x_{i}\right),\Phi \left(x_{j}\right)>\\ & = & \Phi {\left(x_{i}\right)}^{T}\Phi \left(x_{j}\right) \end{array}$$

Using this in Equation (Equation 11), the representation of the output weights and the estimated output of the system can be expressed as:(14)$$\begin{array}{ccc} w & = & \Phi {\left(\Phi^{T}\Phi +\lambda I\right)}^{-1}Y^{T}\\ & = & \Phi {\left(K+\lambda I\right)}^{-1}Y^{T} \end{array}$$
and:(15)$$\begin{array}{ccc} Y(x) & = & Y{\left(K+\lambda I\right)}^{-1}\Phi^{T}\Phi \left(x\right)\\ & = & Y{\left(K+\lambda I\right)}^{-1}\kappa \left(x\right) \end{array}$$
respectively. In this paper, we employ the RRNN and KRR algorithms and the ECAV based feature extractor to build a novel classification system.

## 3. Algorithm for Sea-Land Clutter Classification

### 3.1. Overview of the System

Figure 1 shows the overall organization of the system, consisting of four key steps, namely dataset management, preprocessing, feature extraction, and classification.

The detailed aspects of these steps are as follows:Dataset management: In general, real clutter data are rarely available due to the nature of the study. However, in our case, we obtained the sea clutter from the IPIX radar. The land clutter, however, has to be simulated. This component of the system handles the aspect of simulating the land clutter and mixes with the real sea clutter. More specifically, the land clutter is modeled as the Weibull amplitude compound Gaussian distribution.Preprocessing: To facilitate easier signal processing operations, the radar returns (whether simulated or real) have to be expressed in the quasi-stable state form. This is feasible only if the signals are pre-treated for short time interval information. In this stage, the sea and land clutter data would be separated into a number of short time interval fragments using framing and windowing techniques.Feature extraction: The features of sea and land returns, based on their inherent nature and amplitude statistics, are extracted from the short time interval information. These features can reflect the different properties between the two clutters in the time domain, such as the energy, the zero-crossing rate, the autocorrelation function, and the shape parameters of the distribution.Classification: Two popular machine learning algorithms, namely RRNN and KRR, are used to classify the sea and land clutter. Both algorithms rely on the features that have been extracted above.

In the following subsections, each of these is covered in detail.

### 3.2. Sea and Land Clutter Preparing

In this paper, sea clutter data were acquired from the IPIX (Intelligent PIXel) radar shared by McMaster University, Canada, with a carrier frequency of 9.3 GHz, pulse repetition frequency of 1 kHz, and range resolution of 30 m. The dataset also includes four polarization states. The datasets include a large number of pure sea clutter with targets embedded wherever possible. The dataset also includes pure sea clutter (without any targets). However, the dataset does not include any land clutter [27].

With the dataset not containing land clutter, we addressed this issue by complementing the dataset with land clutter through simulation. However, modeling land clutter is a rather complex process. First, the land clutter is not spatially uniform. This stems from the fact that land surfaces contain natural and man-made structures, such as uniform terrains, farmlands, vegetation, buildings, roads, and skyscrapers. Secondly, the backscatter may vary with the weather conditions. These two, collectively, render any development of a complete theoretical or precise mathematical model for land clutter a difficult process. Otherwise, the theoretical model under certain simplified conditions can effectively assist research and engineering applications. There are two types of land clutter modeling methods, namely the method based on electromagnetic scattering mechanism, such as the Georgia Institutes of Technology (GIT) model, the Ulaby model, and alike models, and the method based on statistics [28]. However, using models based on statistics is easier, efficient, and versatile. For this reason, statistics based methods are considerably common when modeling land clutter. Among these, the distribution of the amplitude can describe the temporal and spatial variation of land clutter. The experimental results mentioned in [28,29] indicated that the corresponding statistical distributions are Weibull, which is a particular case of the compound Gaussian (CG) distribution. More specifically, the Weibull distribution can be used to describe the land clutter amplitude of uniform terrain for L and X band radars, almost mimicking high resolution radars for land clutters

The CG distribution is widely used for high resolution radar echoes. It can be presented as the product of a non-negative random process and a Gaussian process. The former is known as a texture component, while the latter is known as a speckle component [21,22]. Thus,
$$\begin{array}{c} Z\left(n\right)=\sqrt{\tau (n)}x\left(n\right) \end{array}$$
in which:(16)$$x\left(n\right)=x_{I}\left(n\right)+jx_{Q}\left(n\right)$$
where τ(n) and x(n) are the real non-negative and complex zero-mean Gaussian process components, respectively. In [30], the compound Gaussian (CG) model was used to describe clutter returns as a complex process. The real, nonnegative modulation component is τ; hence, τ>0. The corresponding modulating variate has the following probability density function (PDF).
(17)$$f\left(\tau \right)=\frac{1}{\pi \tau^{3}}\int_{0}^{\infty }e^{-\delta p}\cos\theta \cos\left(\delta r^{q}\sin\theta -\frac{r}{2\tau^{2}}\right)dr$$

 where:$$p=r^{\frac{\alpha }{2}},$$
$$q=\frac{\alpha }{p},$$
$$\delta ={\left[\frac{\Gamma (1+2/\alpha )}{2}\right]}^{\frac{\alpha }{2}},$$

 and:$$\theta =\frac{\alpha \pi }{4}.$$
where τ>0, α is the shape parameter, and the range 0<α≤2 guarantees that the generated clutter conforms to the Weibull distribution. δ is the scale parameter, and Γ(·) is the gamma distribution. *r* is the independent variable of the density function.

### 3.3. Preprocessing of Clutter Data

Conventional and common signal processing algorithms, such as the discrete Fourier transform (DFT) and the fast Fourier transform (FFT), are suitable when the signal to be processed is in the steady state. However, radar clutter is often time-varying, particularly when considered across the whole time-span. Thus, direct application of FFT or DFT and relevant techniques cannot be used directly on radar clutter returns. However, the signal can be considered piecewise steady state signals over a number of time segments. In other words, by segmenting the whole time series into a number of segments, each segment represents a (temporary) steady state process, on which the existing techniques can be applied. This property of relative stability, which is referred to as the quasi-steady state of signal in the short time interval, is an important aspect of preprocessing of clutter data, in particular prior to the feature extraction. More specifically, the overall time-series must be segmented into a number of frames so that they can be framed into short time intervals [31].

One of the key aims (apart from establishing the near steady state signal) of windowing is to avoid the feature mutation between consecutive frames. This is achieved by using a frame shift by *s* points apart. Let x(t) be the sampled signal x(t) with a length of *l* time steps. Let *d* be the length of each frame. Then, the signal x(t) can be segmented into *N* frames, where:(18)$$\begin{array}{ccc} N & = & \frac{\left(l-(d-s)\right)}{s}\\ & = & \frac{l-d}{s}+1 \end{array}$$
Furthermore, let there be a windowing function w(t). The discrete signal x(m) can be preprocessed as follows:(19)$$x_{w}\left(m\right)=w\left(t\right)*x\left(m\right)$$
where 0≤m≤l and * is the convolution operator. In other words, Equation (Equation 19) represents the unit impulse response FIR filter. With this, the signal in the *i*-th frame $f(x_{i}\left(k\right))$ is:(20)$$f\left(x_{i}\left(k\right)\right)=w\left(k\right)*x\left(s\left(i-1\right)+k\right)$$
where 1≤i≤N and k=1,2,⋯,d. The selection of frame length *d* plays an important role in the extraction of relevant information. In our case, we determined the appropriate value for *d* using experimental evaluation.

### 3.4. Short Term Time-Domain Signal Processing and Parameter Estimation

Radar returns from sea, land, and targets of interest are always mixed. Although sea-land maps may be of help, as discussed before, it is not always possible to make use of pre-loaded maps, or they may not offer the full capability to distinguish between clutter types and targets. As such, the selection of features that would aid the classification algorithms is of paramount importance. There exists a body of work on the aspects of feature selection for clutter classification. Features of interest include characteristics of the sea clutter, such as the amplitude value, the shape parameter of the distribution, and high order features like kurtosis, skewness, and reflection coefficients. The features we opt for in this paper are based on windowed signals and include the signal short time energy, the zero-crossing rate, the autocorrelation function, and the shape parameters to structure the feature space. We discuss these features in detail below. The simulation results based on the IPIX processing radar datasets in 1998 and simulated land clutter data are demonstrated for the ease of understanding.

Short time energy:The radar echo intensity, which is representative enough of the return energy, can be used to discriminate between the two classes of clutter [22]. From the signal processing perspective, both the short time amplitude function and the energy function can characterize the energy information of the quasi-steady state signals. The latter have a magnifying effect on different signal characteristics. In our work, we extract the time domain short time energy as a distinguishing indicator and compare it with the amplitude feature, as in [2]. The mathematical formulation is as follows: Assume that the short time energy of the signal in the *i*-th frame is E(i). The general formulation of the short time energy of the whole signal x(t) is defined as:
(21)$$\begin{array}{ccc} E[x(t)] & = & \left\{E(1),E(2),\ldots ,E(N)\right\} \end{array}$$
where *N* is the number of frames. Figure 2 shows the raw and short-term signals of the sea-land clutter ((a) and (b)). We also show the distribution of the energy amplitude as a histogram for each of these clutters by binning energy amplitudes accordingly. From the histograms, it can observed that the energy for sea clutter is distributed from $5.5\times 10^{5}$ to $7.0\times 10^{5}$ J, while the energy for the land clutter is distributed from $0.95\times 10^{6}$ to $1.15\times 10^{6}$ J. The range of the distributions can easily facilitate identifying the classes of these clutters.Short time zero-crossing rate: Given a radar system with a predetermined configuration, such as the frequency, polarization, and resolution, the echo from the clutter would vary depending on a number of parameters as discussed before. As for returns from sea and land, it is worth noting that the sea surface is partially homogeneous, while the land surface is heterogeneous with complex terrain conditions, in particular including the discrete scatters such as buildings and other structures. These differences lead to different amplitude profiles for the clutters from sea and land [29]. One of the important metrics for estimating the frequency value of a signal is zero-crossing (ZC) [32], which essentially accumulates a number of times a given signal crosses the zero-line. Albeit that it is simple, it is an effective technique: it not only has much fewer calculation requirements than the traditional fast Fourier transform (FFT) approach, but also demonstrates a high sensitivity to the amplitude change of the signal [32]. In this work, the ZC based feature space is proposed to distinguish the clutter from sea and land. Let C(i) denote the short time zero-crossing rate (STZCR) of the signal in the *i*-th frame, namely $f(x_{i}\left(k\right))$, which is defined as:
(22)$$C\left(i\right)=\frac{1}{2}\sum_{k=0}^{d-1}\left|\mathrm{sgn}[f\left(x_{i}\left(k\right)\right]-\mathrm{sgn}[f\left(x_{i}\left(k-1\right)\right]\right|$$
where 1≤i≤N, 1≤k≤d, and sgn(·) is a symbolic function.The general formulation of the STZCR of the whole signal x(t) is:
(23)$$\begin{array}{ccc} C[x(t)] & = & \left\{C(1),C(2),\cdots ,C(N)\right\} \end{array}$$Figure 3 shows the short time zero-crossing rate and the histogram of the amplitude variation for the sea and land clutter. Although both signals appear to be similar, the distribution of the zero-crossing rate for the sea clutter is more concentrated around 250, when compared against the land clutter. This is primarily due to the homogeneous nature of the sea surface.Short time autocorrelation function: Although radar echoes from sea and ground surfaces pose different challenges to processing, autocorrelation both in the spatial and temporal domain may provide more insight into the behavior of the signals. In this paper, we focus on the autocorrelation in the time-domain. Let A(i) be the autocorrelation function (ACF) of the signal in the *i*-th frame, $f(x_{i}\left(k\right))$. A(i) can be expressed as:
(24)$$A\left(i\right)=\sum_{k=0}^{d-m-1}f(x_{i}\left(k\right))f\left(x_{i}\left(k+m\right)\right)$$
where *d* is the length of the frame and *m* is the delay. The general formulation of the whole signal x(t) is:
(25)$$\begin{array}{ccc} A\left[x(t)\right] & = & \left\{A(1),A(2),\ldots ,A(N)\right\} \end{array}$$Figure 4 shows the result of the autocorrelation for one of the frames (100th frame), of the sea and land clutter signals. We also show the corresponding amplitude histogram for both cases. Although the peaks of the result are the same, the attenuation periods (to zero) for the of sea and land clutter are different.Shape parameter of the clutter distribution: Although the Weibull and K distribution are two different distributions, they both can be derived from the same general form with different values in their parameters [33]. The shape parameter *v* represents the texture of the clutter. Homogeneous clutters lead to high values for *v*, while strong returns such as land clutter are represented by small values [10] of the same. Hence, the shape parameter can be a useful feature in the classification of clutter. The probability density distribution function of different distributions with different shape parameters can be found in [6]. In this paper, we obtained this feature using the moment estimation approach outlined in [5].The general formulation of the shape parameter $\hat{V}\left(x\left(t\right)\right)$ of the whole signal x(t) is:
(26)$$\hat{V}\left[x(t)\right]=\left\{\hat{v}\left(1\right),\hat{v}\left(2\right),\cdots \cdots ,\hat{v}\left(N\right)\right\}$$Figure 5 shows the distribution of the clutters (both sea and land), computed as described above. For the sea clutter, the range of values for the shape parameter is 2.65 to 3.0, while for the land clutter, it is 1.75 to 1.85. The parameter values here reflect the sharpness of the statistical characteristics of the echo under different clutter conditions.

With the four features outlined above, one may build the overall feature space F(x) over an input signal x(t) as follows:(27)$$\begin{array}{ccc} F[x] & = & {\left[\begin{array}{cccc} E(1) & E(2) & \cdots & E(N)\\ C(1) & C(2) & \cdots & C(N)\\ A(1) & A(2) & \cdots & A(N)\\ \hat{V}\left(1\right) & \hat{V}\left(2\right) & \cdots & \hat{V}\left(N\right) \end{array}\right]}_{M\times N} \end{array}$$
where *M* is the number of features of the clutter and *N* is the number of samples, which is the same as the number of frames in the data. The raw inputs are not suitable to be used as features for the RRNN and KRR classifiers. However, the preprocessed version eliminates the dimensionality issues, hence being suitable to be utilized by the RRNN and KRR classifiers.

### 3.5. RRNN Based Classification Algorithms

Both the RRNN and KRR based algorithms outlined above are two, single hidden layer based feed forward neural networks. Given a training dataset *D* = $\left\{x_{i},t_{i}\right\}$, i=1,2,...,N, and activation function g(·), a high-level algorithmic description of the RRNN is given in Algorithm 1, where *H* is the output matrix of hidden layer, $\beta^{*}$ is the hidden-to-output weights, and *w* and *b* are the input-to-hidden weights and bias, respectively. What needs to be emphasized here is that once the node parameters of the hidden layer are randomly generated, they will be fixed during training and solving the output weight with the least-squares approach.
**Algorithm 1** RRNN Algorithm—Training1:$▹\;n_{E}\;:\;\mathrm{Number}\;\mathrm{of}\;\mathrm{epochs}$2:$▹\;n_{B}\;:\;\mathrm{Batch}\;\mathrm{size}$3:▹S:Overalldatasetsize4:▹C:Regularizationparameter5:▹L:Hiddennodenumber6:$M\leftarrow \lceil \frac{S}{n_{B}}n_{E}\rceil$7:Randomlygeneratewb8:**for** 
i = 0; i < M; i++
**do**
9:     H← Equation (Equation 6)10:    **if**
N≥L
**then**11:        $\beta^{*}\leftarrow$ Equation (Equation 8)12:    **else**13:        $\beta^{*}\leftarrow$ Equation (Equation 9)14:    **end if**15:**end for**

## 4. Experimental Results and Analysis

To evaluate the effectiveness of the proposed classification algorithms, we performed a number of experiments in a methodical way. We discuss the procedure and the experimental results in the subsections that follow. We also discuss the results in detail in the relevant subsections.

### 4.1. Experimental Setup

A large number of experiments were carried out to test the effectiveness of the proposed approach. These experiments were:Assessing the impact of the frame length on the classification accuracy by varying the frame length *d*;Understanding the performance of the RRRN classifier through exhaustive evaluation.Understanding the performance of the KRR classifier through exhaustive evaluation.Assessing the impact of different features on the classification accuracy; andAssessing the performance of the RRNN and KRR classifiers against other classifiers.

In all these cases, the data for the tests were as discussed in Section 3, where we mixed the real sea clutter data from the IPIX radar with simulated ground clutter data. In all the experiments, the dataset was split into an 80:20% ratio for training and testing. We also used a modest computing system with an Intel Core-2 processor (San Francisco, CA, USA), clocked at 2.30 GHz with 16 GB RAM. The dataset used for the evaluation in the following subsections contains the sea clutter from an IPIX radar with 60,000 measurements over a region $R_{1}$ and the land clutter, which is modeled as Weibull amplitude compound-Gaussian distribution. We used the same dataset configuration (referred to as DS-0) across all of the experiments below, unless otherwise stated.

### 4.2. Impact of Frame Length on Classification Performance

To understand the impact of frame length on the classification performance, we varied the frame length among {128,256,512,1024}, while using the proposed features, namely the energy, the zero-crossing rate, the autocorrelation, and the shape parameter v (ECAV), for the RRNN and KRR networks. We show the resulting performance in Table 2 and Figure 6. In general, the training time decreased with the frame length, for both methods. For a fixed number of frames, when the frame length increased, the length of the signal decreased, and thus, the resulting feature set became smaller (see Equation (Equation 17)), leading to fewer computations during the classification. Furthermore, the best testing accuracy corresponded to the frame length d=512, for both RRNN and KRR. Although, in theory, this can provide a more detailed description of the signal characteristics, a signal with a limited length leads to fragmented information. The rest of the experiments outlined in the following subsections used a frame length of d=512.

### 4.3. Evaluation of the RRNN Classifier

The overall performance of the RRNN classifier varied with a number of parameters, even when the frame length was fixed. The two main important parameters of interest were the number of nodes in the hidden layer *L* and the regularized parameter *C*, where L∈{100,200,300,400,500,700,1000,1500} and $C\in \{10^{-5},10^{-4},\ldots ,10^{5}\}$. The overall search space was L×C. To understand the overall performance behavior of the RRNN, we performed a grid search over the space of L×C.

The search results are shown in Figure 7. We first show the variation of the performance across the L×C search space (Figure 7a). We then extracted the optimal *L* value, for which the performance was the best and show the variation of the performance across the range of *C* values (Figure 7b). We then repeated the same for the optimal *C* value (Figure 7c).

The results show that a near-optimal performance can be sustained over a large range of the parameter space. In other words, the overall performance was smooth and was insensitive to the variation of the parameters in this range. In the non-smooth region of the performance, the parameter *L* was a dominant factor of the performance for $C\in \{0,\ldots ,10^{5}\}$. Parameters *L* and *C* jointly influenced the overall performance for $C\in \{10^{-4},\ldots ,10^{-1}\}$.

From Equation (Equation 6), the value of the parameter *L* determined the hidden layer output matrix *H*, which in turn influenced the cost error between the estimated output and target output. From Equation (Equation 7), the regularized parameter *C* was a balanced parameter between the training error and the norm of the output weight.

### 4.4. Evaluation of the KRR Classifier

Similar to the RRNN, the overall performance of KRR also depends on the selection of hyperparameters. However, in the case of KRR, the parameters of interest are the width of the Gaussian kernel σ and regularized parameter *C*, where $\sigma \in \{2^{-13},2^{-12},\ldots ,2^{13}\}$ and $C\in \{10^{-5},10^{-4},\ldots ,10^{5}\}$. The overall search space is σ×C, and as before, we performed a grid search on this space.

The Gaussian kernel function used by the KRR is a type of radial basis function, representing the nonlinear feature mapping from the hidden layer to the outputs. The width parameter σ of the KRR controls the radial scope of the function. Carefully adjusting the radial scope can provide a compromise between over-fitting and under-fitting. In addition to σ, KRR obtained a better generalized performance through the regularization parameter *C*.

We show the overall performance results in Figure 8. We first show the variation of the performance across the σ×C search space (Figure 8a). We then extract the optimal σ value, for which the performance was the best, and show the variation of the performance across the range of *C* values (Figure 8b). We then repeat the same for the optimal *C* value (Figure 8c).

As can be observed, the best performance was almost the same as for the RRNN with the relevant parameters. However, the KRR classifier was more sensitive to parameter variation than the RRNN classifier. With sharp, non-smooth variation, the performance of the classifier varied rather sharply and thus indicated that a careful selection of parameters is important for achieving the best possible performance. Although the performance varied rather sharply, improved performance was observed over a subset of the parameter space $\sigma_{0}\times C_{0}$, where $C_{0}\in \left\{10^{0},10^{1},10^{2},10^{3}\right\}$ and $\sigma_{0}\in \left\{2^{2},2^{3},\ldots ,2^{13}\right\}$.

### 4.5. Impact of Features on the Classification Performance

To assess the performance of the proposed feature selection, we evaluated the impact of different features on the overall performance of the RRNN and KRR networks.

Although a large set of features can be engineered, we opted for three common feature sets, extracted through two different wavelets, namely the db3 and fk6 wavelets, and the amplitude, shape parameter *V*, and correlation function (AVC). These features are common in the context of clutter separation and commonly used in signal separation tasks [12,13,34,35,36]. Both the db3 and fk6 wavelets were used to extract four feature vectors from each frame.

Both the RRNN and KRR classifiers were tested on the same dataset, using different features, and the resulting performance is reported in Table 3. It is worth noting that we also compared the performance of the networks with no features specified (marked as “None”). In other words, the networks consumed only the raw data, and no features were specified explicitly. We also illustrate this in Figure 9, showing both the classification performance and training times. As can be observed, the proposed feature set offered the best possible performance. In addition to this, the training time for KRR was better than the RRNN.

### 4.6. Comparison against Other Classifiers

To assess the relative performance of RRNN and KRR classifiers, we used the support vector machine (SVM) and extreme learning machine (ELM) classifiers as the baselines due to [14]. To make the comparison fair, we maintained the same feature set across all these classifiers and maintained d=512.

First, we performed the evaluation over the existing dataset configuration, which was used in all experiments hitherto. We show the overall resulting performance in Figure 10. As can be observed, RRNN offered the best and KRR the second best classification performance. The testing accuracy and training time of RRNN were both better than ELM and SVM.

To further test the overall performance of the classifiers, we introduced two additional dataset configurations, resulting in an exhaustive evaluation of the classifiers over three different dataset configurations, namely:Dataset Configuration-1 (DS-1): sea clutter from an IPIX radar with 120,000 measurements over a region $R_{1}$. The land-clutter is modeled as a Weibull amplitude compound-Gaussian distribution. In other words, the sea-clutter has twice as much as DS-0.Dataset Configuration-2 (DS-2): sea clutter from an IPIX radar with 60,000 measurements over a region $R_{2}$. The land-clutter is modeled as a Weibull amplitude compound-Gaussian distribution.Dataset Configuration-3 (DS-3): sea clutter from an IPIX radar with 60,000 measurements over a region $R_{1}$. The land-clutter is modeled as a log-normal distribution.

We show the overall performance of the RRNN and KRR classifiers against the SVM and ELM classifiers tested over dataset configurations DS-1, DS-2, and DS-3 in Figure 11. As can be observed, the performance of the KRR and RRNN classifiers was almost the same and was better than ELM and SVM in training accuracy, which is because ELM achieves the optimal result by minimizing the cost error between the estimated output and target output, while SVM obtains high-dimensional feature mapping by selecting the appropriate kernel function. Meanwhile both RRNN and KRR consider the trade-off between the output weight and cost error, which improved the classifier’s performance. However, the impact of training a network on an unbalanced dataset is not clear. When comparing the training time of the classifiers, the RRNN offered the best advantage over the others, and it was insensitive to different datasets. However, KRR was sensitive to different datasets, such as a significant increasing in training time under DS-1. Two recent methods, namely SVM and ELM, showed slightly worse accuracy performance than the method we adopted. Despite the training times being similar, this was due to SVM adopting the sequential minimal optimization strategy to solve the quadratic programming problem, and RRNN fixed the random parameters in the hidden layer rather than iteratively updating them to minimize the output error, which is the common pattern of traditional neural networks. We also can see that DS-1 and DS-2 were more suitable for the RRNN, and DS-2 was more suitable for KRR; however, both classifiers’ testing accuracy were reduced under DS-3.

## 5. Conclusions

The identification and separation of clutter from true measurements is one of the essential operations in many target tracking applications. One of the specific problems that is encountered in many target tracking applications is the classification of clutter types, particularly at the interfaces, such as sea-land or air-land, among many others.

Conventionally, these problems are often addressed using model based approaches, such as binary hypothesis testing, or by using well-formulated statistical models of the clutter types. However, with clutter (and target) statistics being highly variable, measurements from radar rarely fit these statistical models. In other words, unless the models are data-adaptive, they are unlikely to perform very well on classification.

Machine learning based techniques are known to be data adaptive and have been used to classify clutter types [2,5,6,8,9,10], often by exploiting certain features of the datasets. In this paper, we proposed two different approaches to this problem, namely the regularized randomized neural network (RRNN) and the kernel ridge regression neural network (KRR), along with a novel feature set, collectively referred to as ECAV, which captures the energy, the zero-crossing rate, the autocorrelation function, and the distribution shape parameter V.

The exhaustive evaluation based on a number of mixed datasets showed that the proposed classifiers offer superior classification performance when used against the proposed feature set and when compared against other classifiers, namely support vector machine and extreme learning machine.

## Figures and Tables

**Figure 1 sensors-20-06491-f001:**
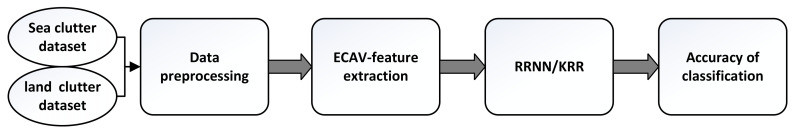
Flowchart of the proposed system.

**Figure 2 sensors-20-06491-f002:**
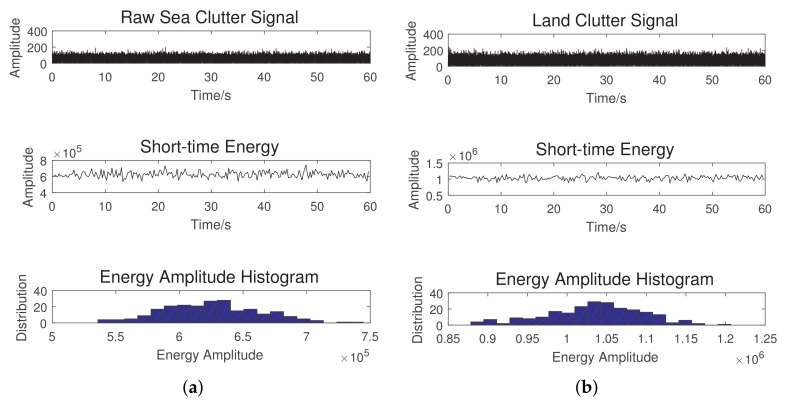
(**a**) Short time energy of the sea clutter; (**b**) short time energy of the land clutter.

**Figure 3 sensors-20-06491-f003:**
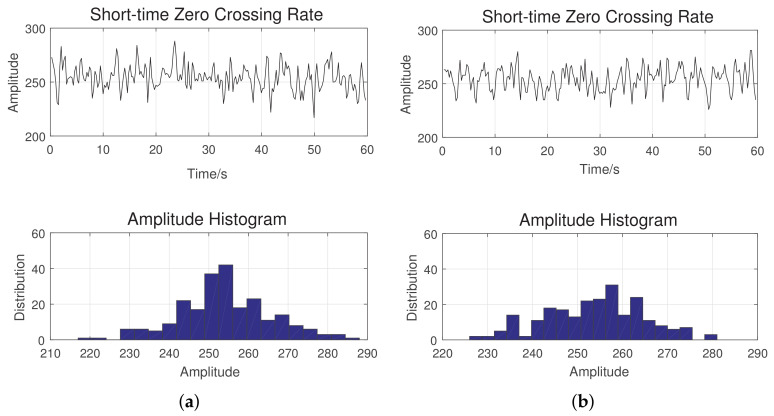
(**a**) Short time zero-crossing rate of the sea clutter; (**b**) short time zero-crossing rate of the land clutter.

**Figure 4 sensors-20-06491-f004:**
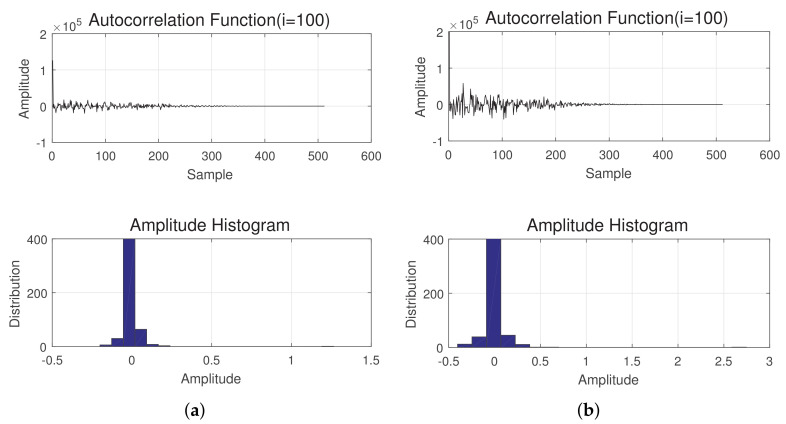
(**a**) Autocorrelation function of the sea clutter (i=100); (**b**) autocorrelation function of the land clutter (i=100).

**Figure 5 sensors-20-06491-f005:**
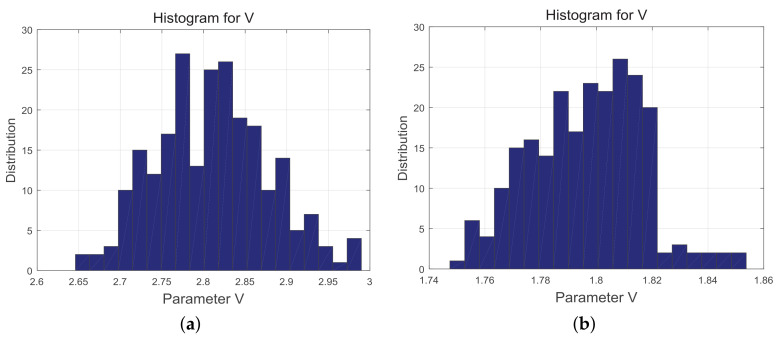
(**a**) Parameter estimation for the sea clutter; (**b**) Parameter estimation for the land clutter.

**Figure 6 sensors-20-06491-f006:**
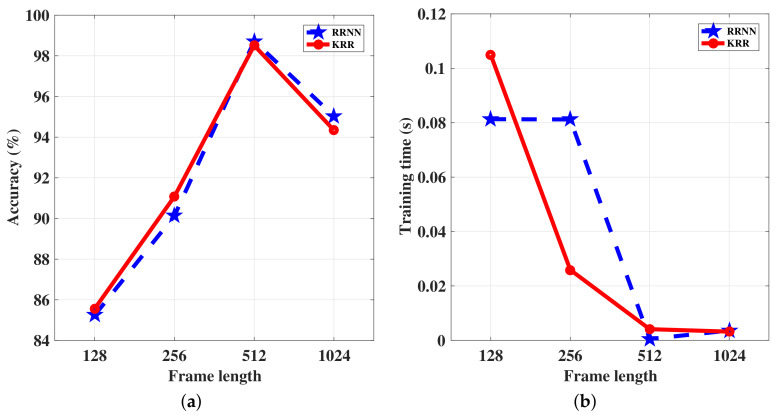
(**a**) Accuracy of the KRR and RRNN algorithms with varying frame lengths; (**b**) Training time for the KRR and RRNN algorithms with varying frame lengths.

**Figure 7 sensors-20-06491-f007:**
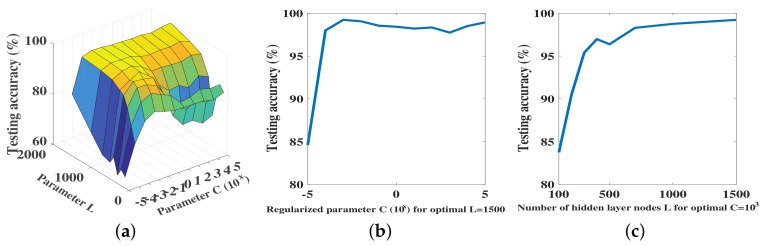
(**a**) Performance of the RRNN across the search space; (**b**) Performance of RRNN for the optimal *L* value; (**c**) Performance of RRNN for the optimal *C* value.

**Figure 8 sensors-20-06491-f008:**
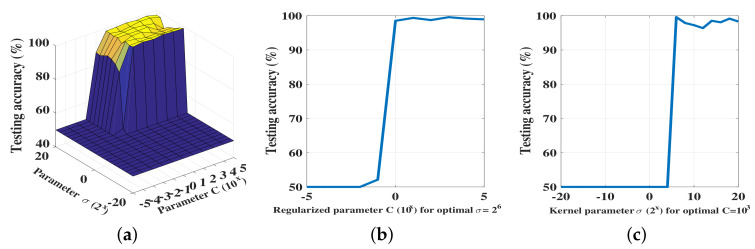
(**a**) Performance of the KRR algorithm across the search space; (**b**) Performance of KRR algorithm for the optimal σ value; (**c**) Performance of KRR algorithm for the optimal *C* value.

**Figure 9 sensors-20-06491-f009:**
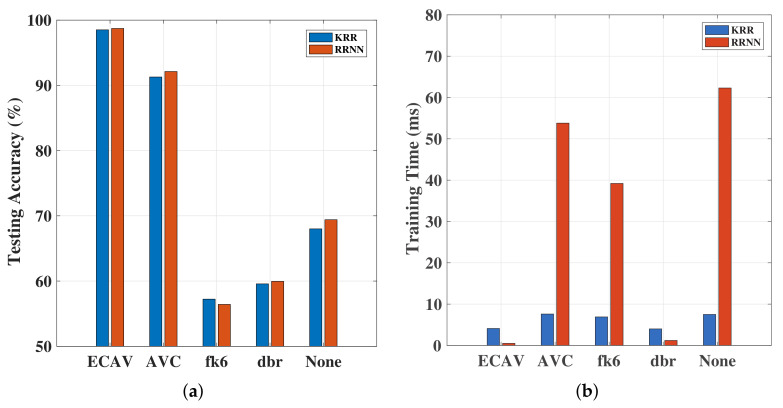
(**a**) Accuracy of the KRR and RRNN algorithms across different feature sets; (**b**) training time for the KRR and RRNN algorithms across different feature sets.

**Figure 10 sensors-20-06491-f010:**
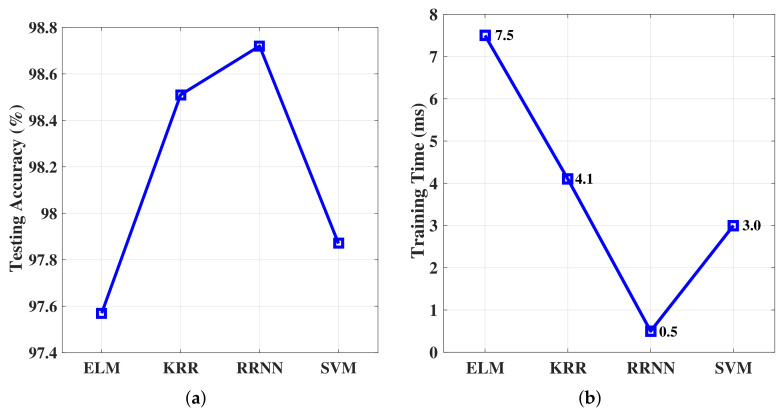
(**a**) Testing accuracy of ELM, KRR, and RRNN based on the ECAV feature set; (**b**) Training time for the ELM, KRR, and RRNN based on the ECAV feature set.

**Figure 11 sensors-20-06491-f011:**
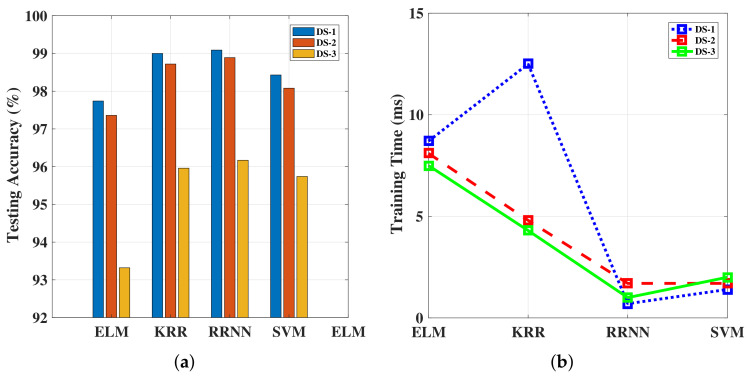
(**a**) Accuracy of the ELM, KRR, and RRNN algorithms across different datasets; (**b**) Training time for the ELM, KRR, and RRNN algorithms across different datasets. DS, dataset.

**Table 1 sensors-20-06491-t001:** Pros and cons of different types of classification solutions.

Solution Type	Method	Pros	Cons
Model-driven	Amplitude analysis, spectrum analysis	Good theoretical foundation, strong interpretability	Difficulty of modeling, parameter estimation
Neural network based data-driven	NN, ANN, ELM	Deep features, independent of clutter model accuracy	Mass sample data requirement, large amount of calculation, weak interpretability
Non-neural network based data-driven	SVM, OCSVM, KNN	Good interpretability, independent of clutter model accuracy	High-dimensional data bottleneck

**Table 2 sensors-20-06491-t002:** Variation of testing accuracy with frame length.

	Accuracy (%)	
**Frame Length** (d)	**RRNN**	**KRR**
128	85.24	85.56
256	90.13	91.08
512	98.72	98.51
1024	95.04	94.34

**Table 3 sensors-20-06491-t003:** Testing accuracy as % (for d=512).

Feature Set	KRR	RRNN
ECAV (Proposed)	98.51	98.72
AVC	91.28	92.12
Wavelet (db3)	59.57	59.95
Wavelet (fk6)	57.23	56.42
None	68.00	69.40

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
