# Peer review of "A Novel Method for Sea-Land Clutter Separation Using Regularized Randomized and Kernel Ridge Neural Networks"

_sensors, 2020, doi:10.3390/s20226491_

Round 1

Reviewer 1 Report

This is an interesting research paper. There are some suggestions for revision.

1. Please use one sentence to discuss existing issues of sea-land clutter classification.

2. In introduction, it discusses different types of classification solutions. Please compare the pros and cons of different types of classification solutions, and list the comparision results in a table.

3. The section 2 is too long. 18 equations are listed. There is no need to list all 18 equations. Please only keep the key contents and add the related references for remaining ones.

4. In section 3, please discuss the special features of sea and land clutter datasets, and explain why Weibull amplitude Compound-Gaussian distribution is suitable for land clutter.

5. As shown in Eq. 20, what is r?

6. As shown in line 251, please explain why tau>0 and 0<alpha<2 and how to get the suitable value of tau and alpha.

7. As shown in Eq. 21, what is s? What (l-(d-s))/s mean?

8. What is k in Eq. 23?

9. In line 328 and 339, Figure number is not shown properly.

10. In section 3.4, where do Figs. 2-5 come from? A lot of contents in section 3.4 are from existing work. Please shorten the discussion. There is no need to list many existing equations.

11. In lines 366-367, it mentions "These features are now ready to be utilized by the RRNN and KRR classifiers". Please explain the corresponding reasons.

12. Section 3.5 only shows RRNN algorithm. Please change the title of section 3.5 or add the formalized KRR algorithm.

13. In algo. 1, what do T, H, beta*, and w mean? What is genNetworkConfig()? What does (s/n_B)* n_E mean? More explantions of algo. 1 are needed.

14. Please analyze and compare the time complexity of the proposed algorithms with existing solutions.

15. Please discuss which datasets are suitable for RRNN and KRR respectively.

16. More recently published solutions need to be discussed.

Reviewer 2 Report

This work tries to separate backscatter signals from sea and land using KRNN. The manuscript is well organized and easy to read.

(1) The reviewer would like to ask for further elaboration of the land clutter simulation. One reference (23) is cited to support the idea of the authors to simulate the land clutter with Compound Gaussian distribution concept. The cited reference, however, is results for L-band radar. The backscatter mechanism can be different for X-band radar (IPIX). Please discuss further followings:

- what is the general understanding of land clutter simulation in the radar community

- validity of the land clutter simulation in the present work

(2) Add reference for IPIX radar

(3) Line 328:

‘Figure ??’

(4) Reference 23:

‘distribution-l-band vv polarisation perspective’ must be ‘distribution - L-band VV polarisation perspective’

Author Response

Dear reviewer,

We would like to thank you for your constructive comments and suggestions that helped to greatly improve the paper.In the following, we are providing a detailed response to each of comments(and you also can see the attachment):

(1) The reviewer would like to ask for further elaboration of the land clutter simulation. One reference (23) is cited to support the idea of the authors to simulate the land clutter with Compound Gaussian distribution concept. The cited reference, however, is results for L-band radar. The backscatter mechanism can be different for X-band radar (IPIX). Please discuss further followings:

- what is the general understanding of land clutter simulation in the radar community

We have addressed this issue by including a text to cover this point. Please see line numbers 235-240 in page 8 .

With the dataset not containing land clutter, we addressed this issue by complementing the dataset with land clutter through simulation. However, modeling land clutter is a rather complex process. First, the land clutter is not spatially uniform. This stems from the fact that land surfaces containing natural and man-made structures, such as uniform terrains, farmlands, vegetation, buildings, roads, and skyscrapers. Secondly, the backscatter may vary with the weather conditions. These two, collectively, render any development on complete theoretical or precise mathematical model for land clutter as a difficult process. Otherwise, the theoretical model under certain simplified conditions can effectively assist research and engineering applications. There are two types of land clutter modeling methods, namely, method based on electromagnetic scattering mechanism, such as Georgia Institutes of Technology (GIT) model, Ulaby model and alike , and method based on statistics [29]. However, using models based on statistics is easier, efficient, and versatile. For this reason, statistics-based methods are considerably common when modelling land clutter.

- validity of the land clutter simulation in the present work

We have addressed this issue by including a text to cover this point. Please see line numbers 239-245 in page 8 .

Among these, the distribution of amplitude can describe the temporal and spatial variation of land clutter.  Experimental results mentioned in [29, 30] indicate that the corresponding statistical distributions are Weibull,  which is a particular case of Compound-Gaussian (CG) distribution. More specifically, Weibull distribution can be used to describe the land clutter amplitude of uniform terrain for L and X band radars, almost mimicking high-resolution radars for land clutters

 (2) Add reference for IPIX radar

We have addressed this issue by adding relevant references (27 and 28).

(3) Line 328:‘Figure ??’

We have addressed this issue by adding the correct figure number. Please see line number 319 in page 11.

(4) Reference 23:

‘distribution-l-band vv polarisation perspective’ must be ‘distribution - L-band VV polarisation perspective’

We have addressed this issue byf fixing the reference as suggested. Please see line number 581-583,page20.

Reviewer 3 Report

The authors presents an interesting method for sea-land clutter separation using two schemes, a regularized randomized neural network (RRNN) and a kernel ridge regression neural network (KRR). My recommendations to the authors are:

  1. Review the redaction of the whole document. 
  2. Review the format of the document, including fonts (for both, text and equations), paragraphs and linespaces to ensure its homogeneity.
  3. Consider to improve the size and quality of the figures throughout the document for a better and easier view. 

Author Response

Dear reviewer,

We would like to thank you for your constructive comments and suggestions that helped to greatly improve the paper.

We believe we have addressed all the issues by carefully revising the entire manuscript and addressing all the issues very carefully.

Round 2

Reviewer 1 Report

All my concerns have been addressed. This paper is ready for publication.